# Insight into the Reaction of Alexidine with Sodium Hypochlorite: A Potential Error in Endodontic Treatment

**DOI:** 10.3390/molecules26061623

**Published:** 2021-03-15

**Authors:** Barbara Czopik, Monika Ciechomska, Joanna Zarzecka, Maciej Góra, Michał Woźniakiewicz

**Affiliations:** 1Jagiellonian University Medical College, Faculty of Medicine, Institute of Stomatology, Department of Conservative Dentistry with Endodontics, Montelupich 4, 30-155 Kraków, Poland; j.zarzecka@uj.edu.pl; 2Laboratory for Forensic Chemistry, Department of Analytical Chemistry, Faculty of Chemistry, Jagiellonian University, Gronostajowa 2, 30-387 Kraków, Poland; monikaciechomska@wp.pl (M.C.); michal.wozniakiewicz@uj.edu.pl (M.W.); 3Department of Organic Chemistry, Faculty of Chemistry, Jagiellonian University, Gronostajowa 2, 30-387 Kraków, Poland; maciej.gora74@gmail.com

**Keywords:** alexidine, sodium hypochlorite, chlorhexidine, root canal treatment, ultra-high-performance liquid chromatography-mass spectrometry, p-chloroaniline

## Abstract

Therapeutic success in endodontic treatment depends on successful infection control. Alexidine dihydrochloride (ALX) was recently proposed as a potential alternative to 2% chlorhexidine (CHX) as it possesses similar antimicrobial properties, expresses substantivity and does not produce p-chloroaniline (PCA) when mixed with sodium hypochlorite (NaOCl). However, the products released in this reaction have not been described to date. The aim of this study was to identify detected chemical compounds formed in the reaction of ALX and NaOCl with the ultra-high-performance liquid chromatography–mass spectrophotometry (UHPLC-MS) method and assess whether precipitates and PCA are formed in this reaction. Solutions of ALX were mixed with the equivalent volume of 2% and 5.25% (*w*/*v*) NaOCl solutions. As control, 2% (*w*/*v*) CHX was mixed with 2% and 5.25% (*w*/*v*) NaOCl. Samples were subjected to the UHPLC-MS analysis. The mixture of ALX and NaOCl resulted in a yellowish precipitate formation, the amount of which depended on NaOCl concentration. Interaction of ALX and NaOCl resulted in the production of aliphatic amines. No PCA was formed when NaOCl was mixed with ALX. However, for the first time, we identified the possible products of the interaction. The interaction between NaOCl and ALX results in the formation of aliphatic amines; therefore, these compounds should not be mixed during endodontic treatment.

## 1. Introduction

One of the main objectives of endodontic treatment is the eradication of infection from root canal systems by chemical and mechanical preparation. It has been widely proved that microbial reduction improves the prognosis of root canal treatment and therefore is the key to achieving endodontic success, as no apical periodontitis will develop without the presence of bacteria [1]. It is a general agreement that final therapeutic success is usually dependent on successful infection control [2].

Sodium hypochlorite (NaOCl), a basic solution used in irrigation protocol due to its unique tissue-dissolving capacities and broad antimicrobial spectrum, is not able to fulfill all criteria of effective chemical preparation, especially regarding smear layer removal and total eradication of microbiota. One-third to one-half of treated root canals remain infected when irrigated with 5.25% NaOCl only [3]. It is therefore suggested that adjunctive chemical solutions be used.

For enhancing the bactericidal effect of NaOCl, 2% chlorhexidine (CHX) was advocated as an additional antimicrobial agent used in irrigation protocol [4]. It is highly effective in killing Gram-positive and Gram-negative bacteria, as well as yeasts [5]. Additionally, CHX possesses the unique feature of a prolonged antimicrobial effect, even after its removal from root canals, which is a clinical phenomenon called substantivity [6,7]. Hence, it is mainly useful as a final irrigant, especially in retreatment cases in which the root canal system is colonized with high proportions of Gram-positive bacteria [8]. CHX administration as a final rinse is particularly important for effective bacteria removal from dentinal tubules, as NaOCl penetration inside them is very small or nonexistent [9]. Therefore, an adjunct synergistic action of NaOCl and CHX against endodontic pathogens would be advised; however, these two chemical compounds cannot be mixed inside the root canal space, as their interaction results in a brown precipitate formation and the release of the toxic compound p-chloroaniline (PCA) [10,11,12]. The insoluble brown-colored precipitate acts as a chemical smear layer reducing the dentinal permeability in the apical third of the root canal [13] and therefore decreases the penetration of the sealer, which results in impaired sealing and causes microleakage. Furthermore, a mixture of CHX and NaOCl should be avoided because of the possibility of color changes in dental structures [14]. Therefore, an alternative solution would be beneficial, one that possesses the advantageous features of CHX and lacks its potential to form precipitates and toxic compounds when mixed with NaOCl. 

Alexidine dihydrochloride (ALX), a biguanide very similar to CHX, was first introduced to dentistry when tested as a mouth-rinse [15]. It possesses chemical properties similar to CHX and was reported to have an even greater affinity for bacterial virulence factors [16]. A solution of 1% (*w*/*v*) ALX was equally effective in *Enterococcus faecalis* removal as 2% (*w*/*v*) CHX when tested on infected bovine dentin blocks [17]. Furthermore, 1% and 2% ALX provide longer antimicrobial substantivity against these pathogens, compared to 0.5% and 2% (*w*/*v*) CHX, respectively [18].

Kim et al. tested ALX as an alternative to CHX as a root canal irrigating solution [19]. In that study, four different concentrations (0.125%, 0.25%, 0.5% and 1% (*w*/*v*)) of ALX were mixed with a 4% (*w*/*v*) solution of NaOCl and analyzed with electrospray ionization mass spectrometry (ESI-MS) to determine whether PCA or precipitates were formed. As a result, the interaction of ALX and NaOCl did not produce PCA or precipitates; however, to date, no steps have been undertaken to identify chemical compounds formed in this reaction. The need for complete identification of products released is crucial for understanding the nature of this interaction and further assessment of the potential toxicity of these chemical compounds. The aim of this study was to determine and identify detected chemical compounds formed in the reaction of ALX and NaOCl with the ultra-high-performance liquid chromatography-mass spectrophotometry (UHPLC-MS) method.

## 2. Results

### 2.1. Mixture of CHX and NaOCl

The mixing of CHX and NaOCl resulted in the formation of a red-brown precipitate. The amount of precipitate varied depending on NaOCl concentration (with constant CHX concentration), but the precipitate was always present even if NaOCl concentration was highly diluted (5.25% (*w*/*v*) NaOCl–H_2_O, 1:99 (*v*/*v*)). Figure 1 shows pictures of obtained precipitate in each mixture.

Previously, some researchers claimed that PCA was not formed after the reaction of CHX with NaOCl [10,11,19], while others claimed that it was [12,20]. As NaOCl is quite a strong oxidative agent, the authors of the present paper hypothesized that PCA is generated under the conditions used. To confirm that statement, the precipitate formed in each mixture (Figure 1) was dissolved in methanol then examined by means of UHPLC-MS. Acquired chromatograms and mass spectra were analyzed and compared with those obtained after the analysis of PCA and CHX standard solutions. 

Under the positive electrospray (ESI+) condition, a PCA molecule gives a characteristic ion: 128.0262 *m*/*z* [M + H]^+^, which was monitored (±0.0050 *m*/*z*) during the analysis of the PCA standard mixture. The retention time of PCA was 0.56 min. The analysis of the precipitate mentioned above revealed the peak at the retention time of 0.56 min, corresponding to the ion of 128.0262 ± 0.0050 *m*/*z*. The UHPLC-MS analysis of diluted supernatant showed no significant analytical signals, which may be attributed to significant ion suppression related to high salt concentration.

### 2.2. Mixture of ALX and NaOCl

When mixing ALX and NaOCl, a yellowish precipitate was formed. Similarly to the CHX–NaOCl reaction, the amount of precipitate depended on the NaOCl concentration (with constant ALX concentration). Correspondingly, the precipitate appeared in each case, even for the lowest NaOCl concentration (5.25% (*w*/*v*) NaOCl–H_2_O 1:99 (*v*/*v*), see Figure 1).

Because of the lack of an aromatic ring (or moieties that could react with NaOCl to form aromatic rings) in the ALX structure, no PCA is formed. However, ALX might be oxidized in a similar way to CHX, resulting in the formation of aliphatic amines. As was assumed, no peak characteristic of PCA was present either on a mass spectrum or on a chromatogram of any analyzed precipitate formed in an ALX–NaOCl mixture. However, some other compounds that could be the products of the reaction between ALX and NaOCl were found. Their presence was predicted on the basis of possible reactions occurring in the mixture and confirmed by identifying these compounds on chromatograms (Figure 2) and corresponding MS spectra (see Appendix A, Appendix A) used for prediction of the molecular formula using the dedicated software (Compass Data Analysis, SmartFormula, Bruker, Bremen, Germany). The postulated chemical structures of identified compounds are presented in Table 1. In some cases, two products are possible as products 2a and 2b, and also 3a and 3b, exhibits in tautomeric forms that can easily interconvert. Product 4 is a dehydrogenated form of N-(diaminomethylidene)-N’’-(2-ethylhexyl)guanidine. However, the site of dehydrogenation remains unknown: it depends on the reaction conditions and cannot be determined by the used method. Product 5 is a dehydrogenated form of ALX; there are four fewer hydrogen atoms in product 5 than in ALX. Similarly to the situation mentioned above, the exact site of dehydrogenation remains unknown. The possible situations are, e.g., formation of double bonds or an aliphatic ring. Moreover, an ALX peak was also observed on all chromatograms of the ALX–NaOCl mixture. The scheme of predicted reactions of ALX with NaOCl (Figure 3) was proposed on the basis of the results of the present experiment and the work of Lüttringhaus et al. [21]. Similarly, as in case of CHX, the UHPLC-MS analysis of diluted supernatant showed no significant analytical signals.

The relative reaction rates of the products depending on the concentration of NaOCl reagent have also been investigated and elaborated as the percentage of a particular peak area among investigated compounds (ALX has been skipped due to significantly higher values; see Appendix A
Appendix A).

## 3. Discussion

Previous studies examining the precipitate formed in the mixture of CHX and NaOCl either confirmed [10,11,19] or denied [12,20] the presence of PCA in the precipitate. The present work asserts that PCA was actually formed in each tested mixture of CHX and NaOCl. PCA induces hemato-, spleno-, hepato- and nephrotoxicity and has stronger cyanogenic potential than aniline [22,23]. Exposure to PCA can result in cyanosis, methemoglobinemia and increased sulfhemoglobin levels, the development of anemia and acute intoxication [22]. From that point of view, the inappropriate root canal disinfection regimen could have a noxious effect. 

Interaction between ALX and NaOCl has been described and discussed previously [19,24,25]. Kim et al. [19] and Jain et al. [24] suggested that ALX can be a potential alternative for CHX solution in endodontic treatment. ALX was reported to have a similar chemical structure and properties and be even more potent in deactivating bacteria virulence factors than CHX. If so, ALX would be particularly useful as the final irrigant in the irrigation protocol, making it the last chemical compound that has contact with root canal dentin. However, the authors of those studies did not identify any product formed in the mixture of ALX and NaOCl [19,24,25]. Moreover, those studies on ALX–NaOCl interaction denied precipitate formation [19,24]. In the present work, it has been affirmed that different aliphatic amines are produced from the reaction of ALX and NaOCl. These amines are formed by breaking bonds in the ALX chain or dehydrogenation caused by NaOCl. Furthermore, the cytotoxicity of postulated products of the ALX and NaOCl reaction has been proved [26], and the toxicity of others needs to be further examined, as compounds with a similar structure we identified have neurotoxic potential [27].

The reaction between ALX and NaOCl should be characterized as a complex system. Nevertheless, based on the literature, it is possible to propose the mechanism of this process. Following the work of Hawkins et al. [28], the reaction mechanism depicted in Figure 4 proposes the formation of 2-ethylhexylamine (2), N-(diaminomethylidene)-N’’-(2-ethylhexyl) guanidine (4) and ALX-4H found in the precipitate. One should also note that there is no substantial trend in formation of products in the precipitate; however, in highly diluted NaOCl, the production of compound 5 seems predominant, while simultaneously the amounts for compounds 1 and 2 decrease (see Appendix A in the Appendix A for details). It may indicate that oxidation at positions a and b (Figure 3) requires a lower concentration of the oxidative agent than in positions a and c. 

For the first time, we have confirmed that the precipitate formed from the reaction of ALX and NaOCl is likewise formed from the reaction of CHX and NaOCl. The color of the precipitate is yellowish, and PCA is not formed. The differences regarding precipitate detection between our study and previous studies [19,24] may be due to different methodologies and experimental designs. Kim et al. [19] and Jain et al. [24] mixed ALX with NaOCl and observed color changes in the reaction solutions in test tubes, which was followed by observation of precipitates on the dentinal surfaces of teeth samples irrigated with an ALX–NaOCl mixture in scanning electron microscopy (SEM) under ×500 and ×1000 magnification. They did not observe precipitate formation on the dentinal surface [19,24]. In the present study, we have confirmed with more advanced analytical techniques that the yellowish precipitate is formed in the reaction of ALX–NaOCl. However, the sole fact of precipitate formation may not impair root canal obturation in the aspect of microleakage promotion, as the occlusion of dentinal tubules was not observed in SEM by other authors [19,24]. Therefore, further testing should be conducted to assess the influence of ALX on dentin properties, dentinal tubule penetration and root canal sealing parameters. Only then can it be concluded that this compound has a better therapeutic value in endodontic treatment than the widely used CHX.

## 4. Materials and Methods

### 4.1. Materials

Methanol and acetonitrile (MS purity grade) were purchased from Sigma-Aldrich (St. Louis, MO, USA). Formic acid and ethanol were supplied by Merck (Darmstadt, Germany). Alexidine dihydrochloride (> 98% purity grade) was purchased from Santa Cruz Biotechnology (Dallas, TX, USA). Solutions used in endodontics protocols, namely Gluxodent (2% (*w*/*v*) solution of chlorhexidine digluconate), CHLORAN 5.25% (5.25% solution of sodium hypochlorite) and CHLORAN 2% (2% (*w*/*v*) solution of sodium hypochlorite), were purchased from Chema-Elektromet (Rzeszów, Poland). 4-Chloroaniline (p-chloroaniline) was supplied by Sigma-Aldrich (St. Louis, MO, USA). Ultrapure water was generated in-lab using a Milli-Q system from Merck Millipore (Darmstadt, Germany).

### 4.2. Preparing of Stock and Standard Solutions

The stock solution of ALX (4 mg mL^−1^ calculated for base compound) was prepared by dissolving analytical standard in a mixture of water and ethanol 4:1 (*v*/*v*). The stock solution of PCA (1 mg mL^−1^) was obtained by dissolving PCA in methanol. The standard working solution of ALX and the nonstandard working solution of CHX (500 ng mL^−1^) were prepared by diluting ALX stock solution or CHX commercial solution used in endodontics protocols in water–acetonitrile 1:1 (*v*/*v*), respectively. Standard solutions of NaOCl were prepared by diluting 5.25% (*w*/*v*) NaOCl solution in water (1:0, 1:1, 1:4, 1:9, 1:19, 1:49, 1:99 (*v*/*v*)). Moreover, a 2% (*v*/*w*) NaOCl commercial solution was used, without previous diluting. The standard solution of PCA (200 ng mL^−1^) was prepared by diluting stock solution in water and acetonitrile 1:1 (*v*/*v*). All solutions were stored in a refrigerator (+4 °C).

### 4.3. Sample Preparation Procedure

Solutions of ALX and CHX were mixed with the equivalent volume (100 μL) of NaOCl standard solutions. A washing procedure was used to remove sodium ions, which can interfere with analytes and form adduct ions in the ion source of the mass spectrometer. Firstly, liquid supernatant from each sample was withdrawn and the precipitate was washed with cold water (+4 °C). Then, the sample was stirred and centrifuged (10 min, 10,000 rpm, 0 °C). The washing procedure was repeated three times. Afterward, the precipitate was dissolved in 500 μL of methanol. Solutions made from a mixture of CHX and NaOCl and a mixture of ALX and NaOCl were 1000-fold and 50-fold diluted, respectively, in water-acetonitrile mixture (H_2_O–ACN) 1:1 (*v*/*v*). Prepared samples were subjected to the UHPLC-MS analysis. 

The liquid supernatant samples were 10,000-fold diluted in H_2_O–ACN 1:1 (*v*/*v*) and subjected to the UHPLC-MS analysis.

### 4.4. UHPLC-MS Analysis

The UHPLC-MS system consisted of the UltiMate 3000 RS liquid chromatography system (Dionex, Sunnyvale, CA, USA) coupled with a tandem mass spectrometer with quadrupole and time-of-flight analyzers (Micro-TOF-Q II, Bruker, Bremen, Germany). A mobile phase consisted of water with 0.1% (*v*/*v*) formic acid (phase A) and acetonitrile with 0.1% formic acid (phase B). In each analysis, 5 μL of sample was injected by an autosampler, and the separation was carried out in isocratic mode (30% A, 70% B) within a Hypersil Gold Phenyl column (50 mm × 2.1 mm I.D., particles 1.9 µm, Dionex, Sunnyvale, CA, USA) for 16 min. The flow rate of the mobile phase was 0.4 mL min^−1^. The column was thermostated at 25 °C. The electrospray ion source was operated under positive ionization conditions (ESI+). The other MS parameters were as follows: nebulizer pressure: 2.0 bar; dry gas flow: 5.5 L min^−1^; drying gas temperature: 200 °C; capillary voltage: −500 V. The accurate mass of each ion was calibrated using sodium formate clusters. SmartFormula (Bruker) was used for the prediction of molecular formulas. The mass range was from 50 to 1500 *m*/*z*.

## 5. Conclusions

The interaction of ALX and NaOCl results in the formation of aliphatic amines. This study showed that the reaction does not result in PCA formation. However, for the first time, we have shown that this interaction results in the formation of a yellowish precipitate, and we identified possible products of the reaction that can be potentially harmful. Therefore, ALX and NaOCl should not be mixed in endodontic irrigation protocol.

## Figures and Tables

**Figure 1 molecules-26-01623-f001:**
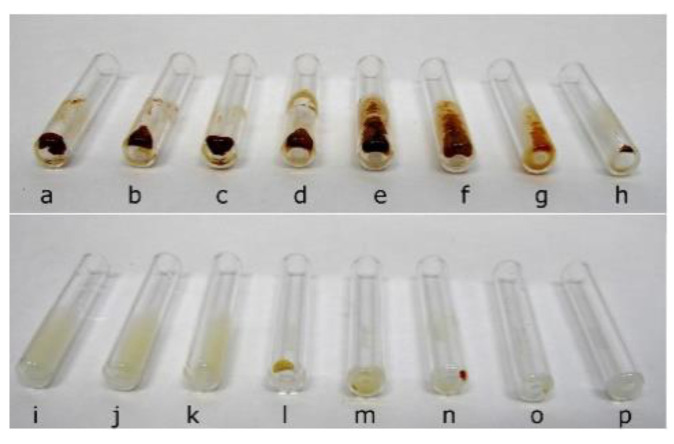
Precipitates obtained after adding chlorhexidine (CHX) to 5.25% sodium hypochlorite (NaOCl) (**a**); 2% (*w*/*v*) NaOCl (**b**); and 5.25% (*w*/*v*) NaOCl:H_2_O (**c–h**) mixed in the ratio of 1:1, 1:4, 1:9, 1:19, 1:49 and 1:99 (*v*/*v*), respectively, and precipitates obtained after adding alexidine dihydrochloride (ALX) to 5.25% (*w*/*v*) NaOCl (**i**); 2% NaOCl (**j**); and 5.25% (*w*/*v*) NaOCl:H_2_O (**k–p**), mixed in the ratio of 1:1, 1:4, 1:9, 1:19, 1:49 and 1:99 (*v*/*v*), respectively.

**Figure 2 molecules-26-01623-f002:**
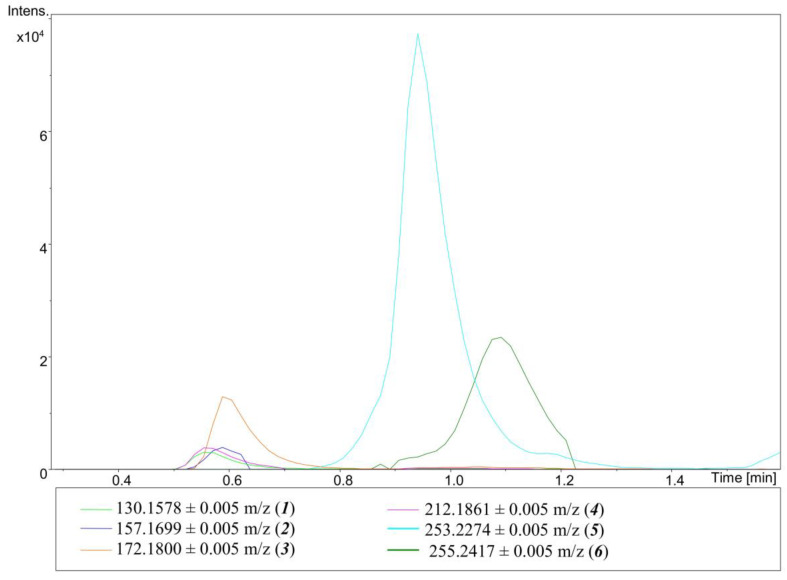
Chromatogram obtained for a mixture of ALX and 5.25% NaOCl–H2O 1:49 (*v*/*v*). The numbers in the brackets correspond with the numbers in Table 1.

**Figure 3 molecules-26-01623-f003:**
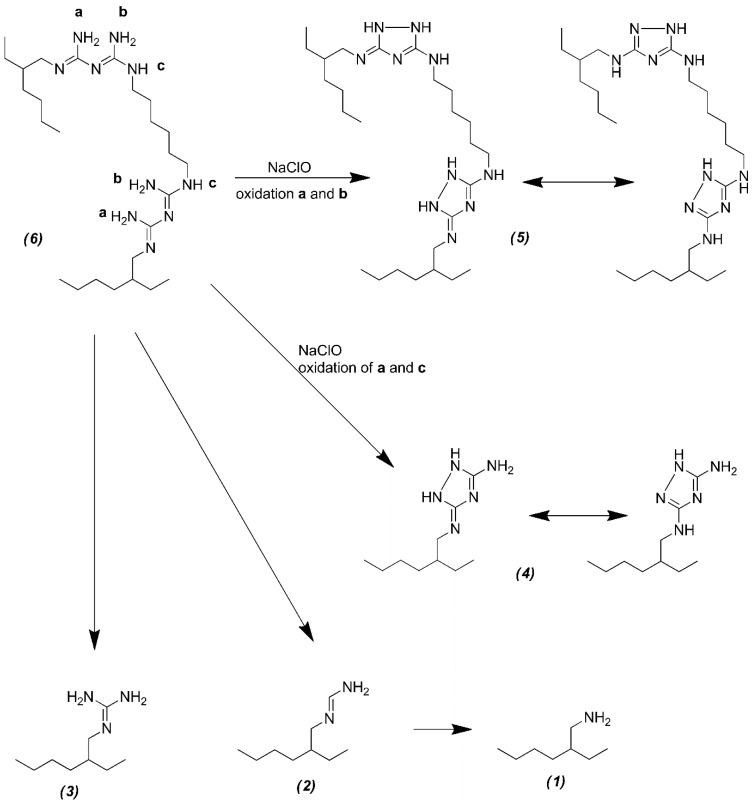
The scheme of possible reactions between ALX and NaOCl (numbers in the brackets correspond with the numbers from Table 1). Letters a, b, c and label nitrogen atoms in the amino groups of guanidine moieties in ALX molecule.

**Figure 4 molecules-26-01623-f004:**
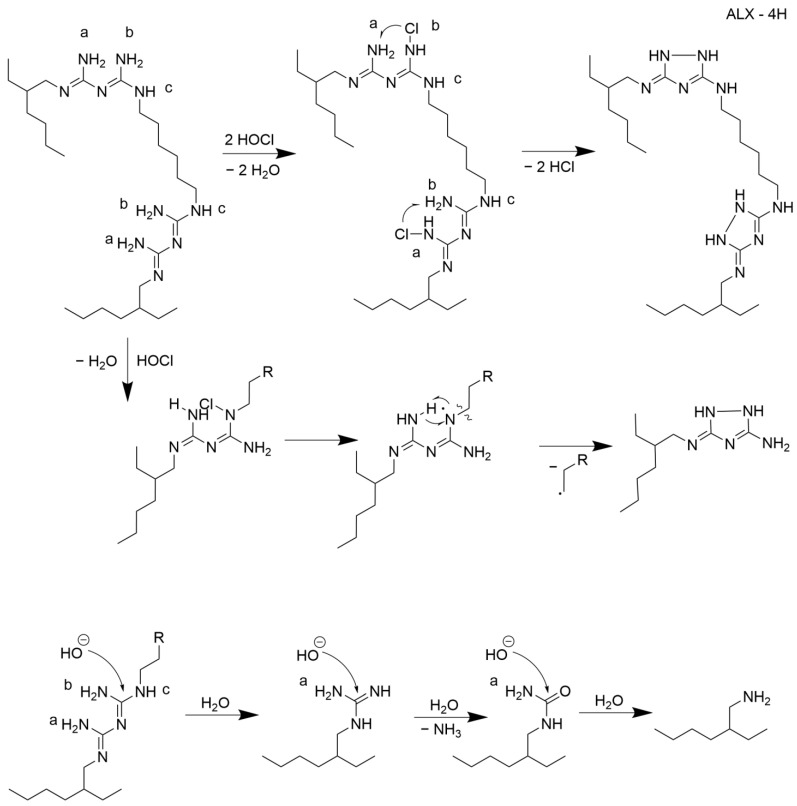
The scheme of reaction mechanism between ALX and NaOCl (the numbers in the brackets correspond with the numbers from Table 1). Letters a, b, and c label nitrogen atoms in the amino groups of guanidine moieties in ALX molecule.

**Table 1 molecules-26-01623-t001:** Compounds identified as generated in the reaction between ALX and NaOCl.

No.	Formula	Predicted Structure	Name	Characteristic Ion*m*/*z*	Retention Time (min)
1	C_8_H_19_N	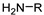	2-ethylhexylamine	130.1578 [M + H]^+^	0.57
2a	C_9_H_20_N_2_	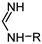	N-(2-ethylhexyl)methanimidamide	157.1694 [M + H]^+^	0.59
2b	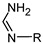	N’-(2-ethylhexyl)methanimidamide
3a	C_9_H_21_N_3_	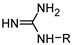	N-(2-ethylhexyl)guanidine	172.1800 [M + H]^+^	0.61
3b	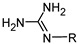	N’’-(2-ethylhexyl)guanidine
4	C_10_H_21_N_5_ *	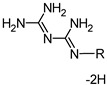	N-(diaminomethylidene)-N’’-(2-ethylhexyl)guanidine	212.1861 [M + H]^+^	0.96
5	C_26_H_52_N_10_ *	ALX-4H	-	253.2274 [M + 2H]^2+^	1.14
6	C_26_H_56_N_10_	**	alexidine (ALX)	255.2417 [M + 2H]^2+^	1.09
R	C_8_H_17_	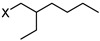	2-ethylhexyl-	113.1330 [M]	-

* More than two structures are possible. ** See Figure 3 for structure (6).

## Data Availability

The data presented in this study are available on request from the corresponding author.

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
