# Peer review of "Insight into the Reaction of Alexidine with Sodium Hypochlorite: A Potential Error in Endodontic Treatment"

_molecules, 2021, doi:10.3390/molecules26061623_

Round 1

Reviewer 1 Report

The paper by Barbara Czopik and colleagues focuses on the reaction of alexidine with sodium hypochlorite in the context of endodontic treatment.

The field of application of the reported research (stomatology) is not the most conventional for a paper in Molecules, but the manuscript is well written and the investigation involves the use of advanced analytical techniques such as ESI-MS and UHPLC-MS. The authors used these methods to investigate the reaction between ALX and NaOCl. My main concern about this version of the manuscript is the overall lack of quantitative data. In my opinion, results should be integrated and I report some suggestions in the following.

The authors state that the reaction did not produce p-chloroaniline. The discussion of the possible reaction mechanisms and of the presumed product is clearly reported (page 3 and Table 1). I would suggest reducing the size of Table 1.

In “Sample preparation procedure”, it is not very clear if the authors completely discarded the supernatant. The content of the liquid should be analyzed to search for compound traces.

Moreover, the paper lacks of discussion of quantitative aspects, which should be investigated by UHPLC-MS. The authors should clearly state and discuss kinetic aspects, since they are relevant in the context of this reaction, and provide more details on the conversion % of the starting compounds to the final products. Finally, a figure showing the most relevant mass spectra should be included.

Author Response

Dear Madam/Sir,

Thank You so much for spending Your time on reviewing our manuscript. We truly appreciate Your comments and suggestions, which all together - with Your level of expertise –  are for us the most valuable source of knowledge. Please find below our proposed answers to Your suggestions.

I allowed myself to organize this response in paragraphs with Your comments in quotations, in order to make it clear and save Your precious time.

  1. “I would suggest reducing the size of Table 1”

Table 1 size was reduced as Reviewer suggested. To accomplish this task we decided to change presentation of chemical structures. 

  1. “In “Sample preparation procedure”, it is not very clear if the authors completely discarded the supernatant. The content of the liquid should be analyzed to search for compound traces.”

Authors would like to apologize for this shortcoming. Authors analyzed the supernatant collected from the sample using LC-MS system, however due to the requirement of multiple sample dilution before the analysis, no significant analytical signals were found. It might be because of the strong ion suppression effect related to high salt concentration in sample (even after dilution). At this stage authors decided to focus on the precipitate only as its composition is of particular relevance for endodontic treatment.

Appropriate amendments were done in sections 2.1, 2.2 and 4.3.

  1. “the paper lacks of discussion of quantitative aspects, which should be investigated by UHPLC-MS. The authors should clearly state and discuss kinetic aspects, since they are relevant in the context of this reaction, and provide more details on the conversion % of the starting compounds to the final products”.

Thank you for this comment. Indeed the quantitative aspects of reaction between ALX and NaOCl are interesting, nevertheless they are out of the scope of this work. Authors would like to emphasize that in the presented research, they decided to focus solely on qualitative analysis of the precipitate which is of high interest in terms of endodontic treatment. We agree that there is a space for further research in this area which will be performed in future, as we are inspired with Reviewer brilliant suggestion.

  1. “Finally a figure showing the most relevant mass spectra should be included”

Thank you so much for this comment. Authors revised the collected results and decided not to include extra mass spectra in the manuscript. Please be so kind to note, that authors used high resolution mass spectrometry which enables us to measure the m/z with high accuracy, which should be considered as efficient in identification of compound composed only from N, O, C. Appropriate measurement results were collected in Table 1 and we felt, that supporting them with additional figures would make some redundancy that we would like to avoid.

Thank You again for Your precious time spent to improve our manuscript and invaluable comments.

If I could in any way answer more questions or this manuscript in Your opinion needs more improvements I’ll be more than happy to change it further.

Kindest Regards,

Barbara Czopik, DDS

Reviewer 2 Report

The article is an experimental study which provides further data about chemical stability of alexidine in the presence of NaOCl. The work is welcome taking into consideration the controversy in the literature and the low number of studies about this subject.

In my opinion the article could be published after a minor revision considering the following issues:

1) Once an acronym introduced it should pe used throughout the text, not a combination of full name and acronym. Ex. chlorhexidine (CHX) etc.

2) The concentration % is % as weigth/weigth or %(w/v)? If it is the last situation, please state this unit of measure as %(w/v) where it is applicable within the text. According to Material and methods part, it seems to be %(w/v). Even if the commercial products which were used are labeled as simple %, please check the supplimentary data provided by the producers.

3) In text:  “4.2 Preparing of stock and standard solutions. Stock solution of alexidine dihydrochloride (4 mg mL-1, ca. 0.5%)”

What does that mean? As mg/mL is an exact concentration and as % is ca. for the same solution? One is concentration as base, the other is as salt or what? Please, clarify in the text for all similar situation.

4) In text: “Standard solutions of CHX and ALX (500 ng mL-1, 0.06%) were prepared by diluting stock solutions in water−acetonitrile (1:1 v/v). Standard solutions of sodium hypochlorite were prepared by diluting 5.25% NaOCl solution in water (1:0, 1:1, 1:4, 1:9, 1:19, 1:49, 1:99 v/v). Moreover, 2% NaOCl standard solution was used, without previous diluting.”

CHX, NaOCl solutions are not standard solutions, are commercial products as authors stated: “Solutions used in endodontics protocols: Gluxodent (2% solution of chlorhexidine digluconate), CHLORAN 5.25% (5.25% solution of sodium hypochlorite) and CHLORAN 2% (2% solution of sodium hypochlorite)”

Please refer as appropriate: “4.2 Preparing of stock and working solutions” and then identify whether is a standard working solution (obtained from a standard substance) or a non-standard working solution (obtained from commercial solutions used in endodontics protocols).

5) In text: “According to the authors best knowledge, interaction between ALX and NaOCl was described only once before [19].”

It is not true. There are more than one article. Please rephrase this and in the Discussion section please take into consideration for discussions at least the following articles about chemical stability of alexidine in the presence of NaOCl.

a) Jain K, Agarwal P, Jain S, Seal M, Adlakha T. Alexidine versus chlorhexidine for endodontic irrigation with sodium hypochlorite. Eur J Dent. 2018 Jul-Sep;12(3):398-402. doi: 10.4103/ejd.ejd_180_17. PMID: 30147406; PMCID: PMC6089054.

b) Alkaline Sodium Hypochlorite Irrigant and Its Chemical Interactions, By: Wright, Patricia P.; Kahler, Bill; Walsh, Laurence J., MATERIALS Volume: ‏ 10 Issue: ‏ 10     Article Number: 1147   Published: ‏ OCT 2017

Author Response

Dear Sir/Madam,

Thank You very much for spending Your time on reviewing our manuscript. We highly appreciate Your comments and suggestions, which all together - with Your level of expertise –  are for us the most valuable source of knowledge. Please find below our proposed answers to Your suggestions. I allowed myself to quote Your review statements in points while replying on them. I hope it will make my response more organized and therefore will allow You to spend less of Your precious time while reading it.

  1. “Once an acronym introduced it should pe used throughout the text, not a combination of full name and acronym. Ex. chlorhexidine (CHX) etc.”

The changes were made throughout whole manuscript as Reviewer suggested.

  1. “The concentration % is % as eight/ eight or %(w/v)? If it is the last situation, please state this unit of measure as %(w/v) where it is applicable within the text. According to Material and methods part, it seems to be %(w/v).”

Authors would like to thank for this remark. The concentration units is obviously w/v and this amendment was introduced throughout the manuscript. We apologize for this mistake.

  1. “In text: “4.2 Preparing of stock and standard solutions. Stock solution of alexidine dihydrochloride (4 mg mL-1, ca. 0.5%). What does that mean? As mg/mL is an exact concentration and as % is ca. for the same solution? One is concentration as base, the other is as salt or what? Please, clarify in the text for all similar situation.”

Authors would like to thank for this remark. The concentration units is mg/mL calculated for base. The concentration in percentage has been deleted. 

  1. “Please refer as appropriate: “4.2 Preparing of stock and working solutions” and then identify whether is a standard working solution (obtained from a standard substance) or a non-standard working solution (obtained from commercial solutions used in endodontics protocols).”

 Thank You so much for this remark and we want to apologize as indeed there were some mistakes in this part. To clarify the protocol, authors indicate that ALX standard working solutions were obtained from standards while CHX non-standard working solution was used prepared from commercial solutions used in endodontic irrigation protocols.

  1. “In text: “According to the authors best knowledge, interaction between ALX and NaOCl was described only once before [19].” It is not true. There are more than one article. Please rephrase this and in the Discussion section please take into consideration for discussions at least the following articles about chemical stability of alexidine in the presence of NaOCl.”

The “Discussion” paragraph was revised and the statement was rephrased into: “Interaction between ALX and NaOCl has been described and discussed before [19, 24, 25]. Kim et al. [19] and Jain et al. [24] suggested that ALX can be a potential alternative for CHX solution in endodontic treatment”. Manuscripts, that were listed by Reviewer were introduced into “Discussion” section and were listed in References:

  1. Jain K, Agarwal P, Jain S, Seal M, Adlakha T. Alexidine versus chlorhexidine for endodontic irrigation with sodium hypo-chlorite. Eur J Dent. 2018;12:398-402.
  2. Wright PP, Kahler B, Walsh LJ. Alkaline Sodium Hypochlorite Irrigant and Its Chemical Interactions. Materials 2017;10:1147.

 “Discussion” paragraph was rewritten with the highlight on the differences in methodology of ALX/NaOCl interaction investigation between those studies and our experiment and how it may impact the quality of root canal obturation and therefore clinical success of the treatment.

Thank You again for Your precious time spent to improve our manuscript and invaluable comments.

I feel honored.

If I could in any way answer more questions or this manuscript in Your opinion needs more improvements I’ll be more than happy to change it further.

Kindest Regards,

Barbara Czopik, DDS

Round 2

Reviewer 1 Report

The authors modified some parts of the manuscript but did not consider the suggestions concerning the analytical part. It should be considered that Molecules is a chemistry-oriented journal and presented data should be supported by adequate method description, discussion and supplementary data/figures.

Author Response

Dear Madam/Sir

Authors would like to thank to the Reviewer for the remarks. The analytical part of the manuscript has been upgraded by adding mass spectra of the compounds (Supplementary materials). Moreover: we changed the manuscript in terms of introducing some consideration about the influence of the concentration of the oxidative agent on the production of particular compounds (Sections 2.2, 3 and supplementary materials).

However, the deeper quantitative analysis of the investigated process is unquestionably interesting, unfortunately  during the experimental part of the study authors did not have the access to appropriate standards or they were beyond economic reasons.

Authors would like to continue this research in future to look more deeply into this problem, taking into account the Reviewer kind remarks.

Thank You so much again for Your level of expertise, time spent of reviewing our research and comments. We highly appreciate it.

If  we could in any way further improve the manuscript, we will be happy to do so.

Kindest Regards,

Barbara Czopik

Michal Wozniakiewicz

Round 3

Reviewer 1 Report

The authors should good will and undertook the effort of introducing mass spectra analysis and comments on quantitative aspects of oxidative agents. Thus, the paper is now a more complete chemistry-oriented research work.